# Hi-Agent: Hierarchical Vision-Language Agents for Mobile Device Control

## Abstract

Building agents that autonomously operate mobile devices has attracted increasing attention. While Vision-Language Models (VLMs) show promise, most existing approaches rely on direct state-to-action mappings, which lack structured reasoning and planning, and thus generalize poorly to novel tasks or unseen UI layouts. We introduce **Hi-Agent**, a trainable hierarchical vision-language agent for mobile control, featuring a high-level reasoning model and a low-level action model that are jointly optimized. For efficient training, we reformulate multi-step decision-making as a sequence of single-step subgoals and propose a foresight advantage function, which leverages execution feedback from the low-level model to guide high-level optimization. This design alleviates the path explosion issue encountered by Group Relative Policy Optimization (GRPO) in long-horizon tasks and enables stable, critic-free joint training. Hi-Agent achieves a new State-Of-The-Art (SOTA) **87.9%** task success rate on the Android-in-the-Wild (AitW) benchmark, significantly outperforming prior methods across three paradigms: prompt-based (AppAgent: 17.7%), supervised (Filtered BC: 54.5%), and reinforcement learning-based (DigiRL: 71.9%). It also demonstrates competitive zero-shot generalization on the ScreenSpot-v2 benchmark. On the more challenging Android-World benchmark, Hi-Agent also scales effectively with larger backbones, showing strong adaptability in high-complexity mobile control scenarios.

## 1 Introduction

Creating intelligent agents capable of assisting users with automated mobile device operations has received growing attention in recent years (Li et al., 2024). The rise of large-scale foundation models (Devlin et al., 2019; Radford et al., 2018; Raffel et al., 2020; Ouyang et al., 2022; Touvron et al., 2023), particularly vision-language models (VLMs) (Lu et al., 2019; Radford et al., 2021; Liu et al., 2023; Bai et al., 2023; Wang et al., 2024a), has opened new possibilities for instruction following, commonsense reasoning, and zero-shot generalization in this domain.

Current methods for building mobile agents are broadly classified by their optimization strategy into two categories: *prompt-based* and *post-trained agents*. Prompt-based approaches leverage powerful, frozen large models through carefully designed prompts and tool-usage workflows (Zhang et al., 2023; Wang et al.; Chen et al., 2024). While demonstrating strong initial capabilities, they are limited by high inference costs and an inability to adapt their parameters to downstream tasks. In contrast, post-trained agents fine-tune smaller, more efficient VLMs via supervised fine-tuning (SFT) or reinforcement learning (RL) for greater adaptability (Bai et al., 2024; Zhang & Zhang, 2024; Qin et al., 2025). Our work focuses on this RL-based post-training approach for mobile device control.

Within the post-trained paradigm, model architecture is a critical design choice. As illustrated in Figure 1, many agents adopt a flat architecture (Figure 1(a)). Some attempt to learn a direct state-to-action mapping, but this brittle mapping struggles to generalize to unseen tasks (Bai et al., 2024). Others employ a single model for both reasoning and decision-making, but this approach often demands massive computational resources and extensive high-quality data for training (Gu et al., 2025). Recently, hierarchical architectures have emerged (Figure 1(b)) to decompose the problem by using a high-level model for reasoning and a low-level model for execution, thereby simplifying the optimization challenge (Agashe et al., 2025). However, the high-level model often remains frozen, precluding true end-to-end learning and co-adaptation between the two levels.

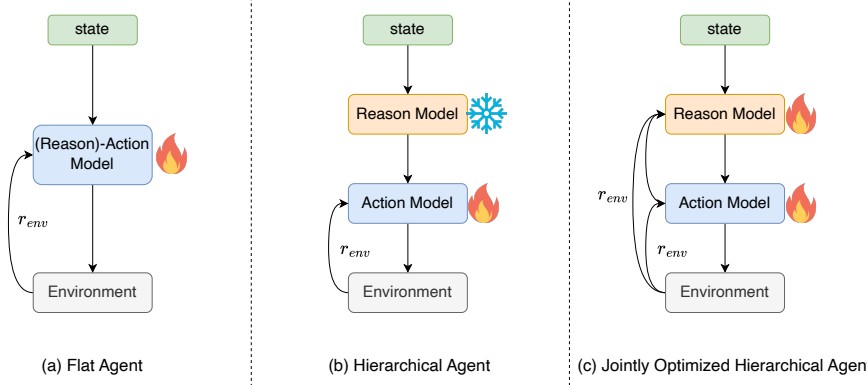

Figure 1: **Different paradigms for mobile control agents.** (a) **Flat Agents** use a single trainable model for state-to-(reason-)action mapping. (b) **Hierarchical Agents** use a planner to improve reasoning, but it is typically a frozen black-box. (c) **Hi-Agent (Ours)** enables **joint optimization** where the high-level reasoning and low-level action models are co-adapted and fully trainable.

To overcome these limitations, we propose a third architectural paradigm: a **jointly optimized hierarchical agent** (Figure 1(c)). We introduce **Hi-Agent**, a hierarchical agent where both the high-level reasoning model ($\pi_h$) and the low-level action model ($\pi_\ell$) are trainable and co-adapted during post-training. This approach marries the structural robustness of a hierarchy with the adaptability of end-to-end optimization, allowing the planner to learn what constitutes an effective subgoal based on direct feedback from the executor's performance.

We introduce a novel training strategy based on Group Relative Policy Optimization (GRPO) (Shao et al., 2024; Guo et al., 2025). To make GRPO tractable for long-horizon tasks, we first reformulate them into a sequence of single-step subgoal predictions, reducing the optimization complexity from exponential ($G^n$) to linear ($n \cdot G$). Second, we introduce a *foresight advantage function* that propagates low-level execution feedback to guide the high-level optimization. This enables stable, critic-free, and sample-efficient joint training.

Our main contributions are as follows:

- We propose **Hi-Agent**, a trainable hierarchical agent with a jointly optimized planner and executor that combines structured reasoning with end-to-end adaptation for mobile control.

- We develop a GRPO-based training framework with a foresight advantage function, which overcomes the path explosion and enables stable credit assignment for high-level planning.

- Hi-Agent achieves SOTA performance and strong generalization, demonstrating robustness, versatility, and scalability across benchmarks like AitW, ScreenSpot-v2, and AndroidWorld.

Experiments show that **Hi-Agent** achieves a new state-of-the-art **87.9%** task success rate on the Android-in-the-Wild (AitW) benchmark, significantly outperforming prior methods. It also demonstrates competitive zero-shot generalization on the ScreenSpot-v2 benchmark and scales effectively on the more complex AndroidWorld benchmark, highlighting its excellent adaptability.

## 2 RELATED WORK

### 2.1 VISION-LANGUAGE AGENTS WITH TOOL-AUGMENTED MOBILE CONTROL

Large vision-language models, augmented by specialized tools, have demonstrated strong performance on various tasks (Yang et al., 2023; , FAIR; Qian et al., 2023). In mobile device control, approaches combine models, tools, and skills to enhance automation. For example, AppAgent (Zhang et al., 2023) builds on GPT-4V by leveraging Android XML files for on-screen localization and learns to use new applications via path exploration or human demonstrations. MobileAgent (Wang et al.) uses a visual module to locate screen elements without XML data, paired with incremental self-planning to traverse app interfaces. Mobile-Agent-v2 (Wang et al., 2025) introduces a multi-

agent paradigm, combining a language model and a vision-language model to support task progression and content-focused navigation. While these methods leverage powerful base models and sophisticated tool coordination, they typically avoid updating the base model's parameters. As a result, performance gains are limited, and the large size of these models—often exceeding hundreds of billions of parameters—can hinder real-world deployment.

## 2.2 Parameter-Efficient Learning for Mobile Device Control

To balance model size and efficacy, researchers have explored fine-tuning vision-language models on demonstration data for mobile device control. Auto-GUI (Zhang & Zhang, 2024) interacts directly with user interfaces—without relying on external tools or low-level system data—by applying gradient-based updates on expert demonstration datasets. DigiRL (Bai et al., 2024) adopts a two-stage reinforcement learning pipeline: it first pretrains a policy in an offline RL setting, then transitions to online RL to refine state-action mappings. DigiQ (Bai et al., 2025a) eliminates the need for online interaction by learning a VLM Q-value function solely from offline data, using temporal-difference (TD) learning on frozen intermediate layers instead of retraining the entire model—achieving performance comparable to DigiRL. However, because these methods directly map tasks to actions, small deviations from the training distribution (e.g., shifts in application locations or UI layout changes) can break the learned mapping and require retraining. Our work addresses this limitation by introducing a reasoning component that transforms direct mappings into a hierarchical "reason first, then act" framework, improving generalization and interpretability.

## 2.3 Reinforcement Learning-based Post-training for Vision-Language Models

Post-training typically refers to applying reinforcement learning (RL) directly to foundation large language models (LLMs) or VLMs without relying on supervised fine-tuning (SFT) as a prerequisite. OpenAI O1 (OpenAI, 2024b) has demonstrated that RL-driven post-training can effectively enhance the reasoning capabilities of LLMs in a scalable manner, requiring fewer computational resources than SFT. To further reduce RL training overhead, DeepSeekMath (Shao et al., 2024) employs GRPO, eliminating the need for a critic model comparable in size to the policy. Instead, it uses group-based rewards to estimate advantages, yielding significant improvements in mathematical, programming, and scientific reasoning tasks. Adapting GRPO to mobile multi-modal control presents two challenges: the exponential growth of reasoning paths and the lack of dense reward signals for high-level planning. We address both issues by designing a hierarchical optimization framework that reduces the reasoning complexity from $G^n$ to $n \cdot G$, and by incorporating a foresight advantage function to guide high-level policy updates using low-level execution feedback.

## 3 Preliminaries

We model mobile device control as a multi-step decision-making process under a Markov Decision Process (MDP), defined as:

$$M_{\text{Interact}} = (\mathcal{S}, \mathcal{A}, \mathcal{P}, \mathcal{R}, \gamma),$$

where $\mathcal{S} = \mathcal{X}_{\text{img}} \times \mathcal{L}$ is the state space of screen images and task instructions, $\mathcal{A}$ denotes atomic UI actions (e.g., click, swipe), $\mathcal{P}$ captures environment transitions, and $\mathcal{R}$ provides task feedback.

While the environment operates at the level of discrete UI actions, subgoal and action generation by language models unfolds token by token. To support RL training over such autoregressive outputs, we follow standard practice (Ouyang et al., 2022) and define a token-level MDP:

$$M_{\text{Gen}} = (\mathcal{S}_{\text{tok}}, \mathcal{A}_{\text{tok}}, \mathcal{P}_{\text{tok}}, \mathcal{R}_{\text{tok}}, \gamma),$$

where $\mathcal{S}_{\text{tok}}$ is the space of sequences, $\mathcal{A}_{\text{tok}}$ is the vocabulary, and $\mathcal{P}_{\text{tok}}$ appends tokens deterministically. Rewards $\mathcal{R}_{\text{tok}}$ are assigned post-generation, based on alignment with oracle actions.

This dual-MDP formulation enables structured learning: we optimize token-level generation via reinforcement learning while evaluating policies in the full multi-step environment.

# 4 METHOD

To address the brittleness of direct state-to-action mappings, our key insight is to introduce dedicated reasoning and action components, transforming this flat mapping into a hierarchical decision process that follows the principle of *first reason, then act*. We define the overall policy as $\pi = (\pi_h, \pi_\ell)$, where the high-level reasoning model $\pi_h$ predicts a semantic subgoal, and the low-level action model $\pi_\ell$ executes the atomic action based on the subgoal and the current screenshot.

We organize this section into three parts. Section 4.1 formalizes our hierarchical structure using recursive value modeling. Section 4.2 introduces our hierarchical post-training method inspired by this decomposition. Section 4.3 presents the data generation pipeline and training implementation.

## 4.1 HIERARCHICAL TASK DECOMPOSITION FOR MOBILE CONTROL

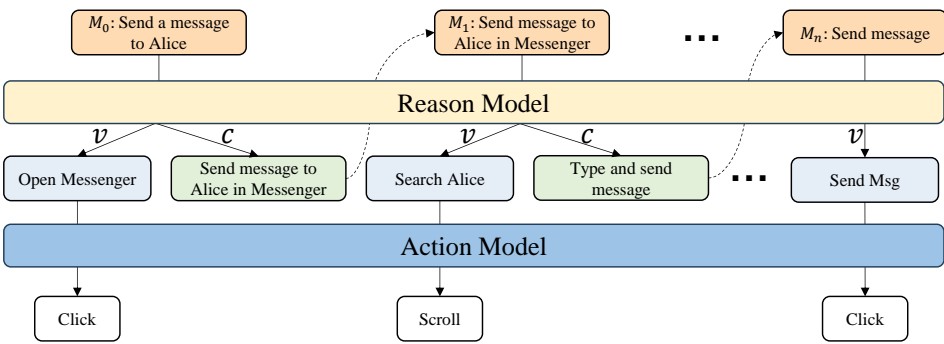

Figure 2: Illustration of recursive task decomposition under a hierarchical policy.

Mobile device control tasks often exhibit natural hierarchical structure. For example, consider the instruction *"Send a message to Alice"*. As shown in Figure 2, this task can be broken down into subtasks such as *"Open Messenger"* and *"Send message to Alice in Messenger"*, the latter of which may be further decomposed into *"Search Alice"*, *"Compose message"*, and *"Press send"*. Each subgoal contributes to completing the overall task and fits into a recursive hierarchy.

In hierarchical RL (Pateria et al., 2021), the recursive structure is often formalized via value function decomposition. Following prior work (Dietterich, 2000; Ghavamzadeh & Mahadevan, 2007), we model the overall task as an MDP $M_{\text{Interact}}$, which captures environment dynamics and UI-level feedback, and decompose it into subtasks $\{M_0, M_1, \ldots, M_n\}$, where $M_0$ is the root. The value function $V_i^\pi(s)$ for subtask $M_i$ under policy $\pi$ is defined as:

$$V_i^\pi(s) = \begin{cases} Q_i^\pi(s, \pi(s)) = V_g^\pi(s) + C_i^\pi(s, g) & \text{if } i \text{ is composite,} \\ \sum_{s'} P(s' \mid s, i) \cdot R(s' \mid s, i) & \text{if } i \text{ is primitive,} \end{cases} \tag{1}$$

where $g = \pi(s)$ is the selected subtask and $C_i^\pi(s, g)$ denotes the expected return after $g$ completes:

$$C_i^\pi(s, g) = \sum_{s', N} P_i^\pi(s', N \mid s, g) \cdot \gamma^N Q_i^\pi(s', \pi(s')), \tag{2}$$

where $(s', N)$ denotes the resulting state and duration after completing $g$. This recursive decomposition provides intuitive motivation that each subtask $M_i$ is optimized not only for its immediate executability (captured by $V_g^\pi(s)$), but also for its long-term impact on overall task success (modeled by $C_i^\pi(s, g)$).

In practice, rather than maintaining a separate policy for every subtask, we implement a compact two-level architecture: a high-level reasoning policy $\pi_h$ that emits semantic subgoals $g_t$, and a low-level action policy $\pi_\ell$ that executes each subtask via atomic actions $a_t$. This design enables cross-task skill reuse and efficient end-to-end training. We further analyze its optimality in Appendix A.

## 4.2 Hierarchical Policy Post-training

While recursive value decomposition offers useful intuition, explicitly modeling value functions $(V_g^\pi, C_i^\pi)$ is computationally expensive and unstable, particularly for LLMs with sparse rewards. We thus adopt Group Relative Policy Optimization (GRPO) (Shao et al., 2024), a scalable, critic-free alternative that computes relative advantages over $G$ sampled outputs from the generation MDP $M_{\text{Gen}}$. The corresponding surrogate objective is:

$$\mathcal{J}_{GRPO}(\theta) = \mathbb{E}_{q \sim P(Q), \{o_i\}_{i=1}^G \sim \pi_{\theta_{old}}(O|q)} \left[ \frac{1}{G} \sum_{i=1}^G \frac{1}{|o_i|} \sum_{t=1}^{|o_i|} \left\{ \right. \right.$$

$$\left. \left. \min \left( \frac{\pi_\theta(o_{i,t}|q, o_{i,<t})}{\pi_{\theta_{old}}(o_{i,t}|q, o_{i,<t})} \hat{A}_{i,t}, \text{ clip} \left( \frac{\pi_\theta(o_{i,t}|q, o_{i,<t})}{\pi_{\theta_{old}}(o_{i,t}|q, o_{i,<t})}, 1 - \epsilon, 1 + \epsilon \right) \hat{A}_{i,t} \right) \right\} \right].$$

$$(3)$$

Here, $\pi_\theta$ and $\pi_{\theta_{old}}$ are the current and reference policies, $q$ is the task input, $o_{i,t}$ is the $t$-th token in output $o_i$ sampled from $\pi_{\theta_{old}}$, $\hat{A}_{i,t}$ is the estimated advantage, and $\epsilon$ is the clipping threshold.

However, applying GRPO to long-horizon tasks presents two major challenges: (1) sampling complexity grows exponentially with trajectory length, requiring $G^n$ rollouts for $n$-step; (2) high-level subgoals are abstract and not directly executable, making reward assignment difficult. To address these issues, we make three key modifications (Figure 3): (1) we decompose $n$-step tasks into $n$ single-step subtasks, reducing sampling cost from $G^n$ to $n \cdot G$; (2) we introduce a foresight reward for each subgoal $g_t$ from $\pi_h$, integrating execution feedback and subgoal quality; (3) we adopt an alternating optimization scheme for $\pi_h$ and $\pi_\ell$ to enable mutual adaptation during training.

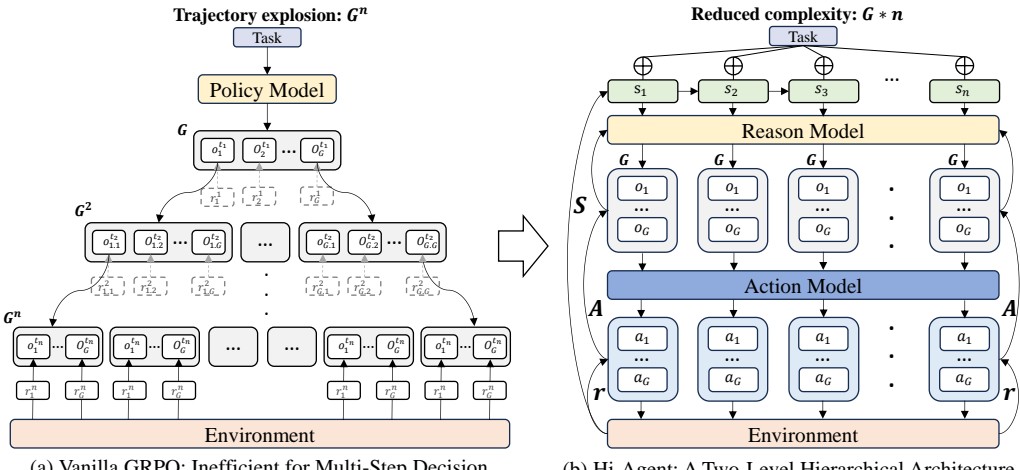

(a) Vanilla GRPO: Inefficient for Multi-Step Decision

(b) Hi-Agent: A Two-Level Hierarchical Architecture

Figure 3: **Hierarchical Policy Optimization. (a)** Standard GRPO incurs exponential sample complexity ($G^n$) and lacks intermediate reward signals in long-horizon tasks. **(b)** Hi-Agent reduces complexity to $n \cdot G$ by decoupling subgoal generation from execution, and enables efficient joint training through foresight-guided subgoal evaluation.

**High-Level Policy Optimization.** At timestep $t$, $\pi_h$ generates a semantic subgoal $g_t$. Inspired by the recursive decomposition (Section 4.1), we design a foresight reward function that encourages $g_t$ to be both immediately executable and conducive to long-term task progress.

To capture both aspects, we combine three reward components: the format reward $r_{\text{fmt}}(g_t)$ is a binary indicator that checks whether $g_t$ conforms to the required schema `<reasoning>...</reasoning><instruction>Instruction:...</instruction>`; the environment feedback reward $r_{\text{env}}(s_t, g_t, a_t)$ evaluates whether the predicted atomic action $a_t = \pi_\ell(s_t, g_t)$ matches the oracle action $\hat{a}_t$ within a tolerance $\epsilon$:

$$r_{\text{env}}(s_t, g_t, a_t) = \mathbb{1} \left\{ \text{type}(a_t) = \text{type}(\hat{a}_t) \wedge \|\text{coord}(a_t) - \text{coord}(\hat{a}_t)\|_2 < \epsilon \right\};$$

and the feasibility reward $\hat{V}_{\text{judge}}(s_t, g_t)$ is evaluated by a frozen vision-language model, instantiated as Qwen2.5-VL-72B-Instruct(Bai et al., 2025b). This model plays the role of an LLM-based

judge (Zheng et al., 2023), estimating whether $g_t$ semantically meaningful and likely to contribute to long-term task success.

These components are combined into a weighted foresight reward:

$$r_t^{(h)} = \lambda_1 \cdot r_{\text{fmt}}(g_t) + \lambda_2 \cdot r_{\text{env}}(s_t, g_t, \pi_\ell) + \lambda_3 \cdot \hat{V}_{\text{judge}}(s_t, g_t), \quad \hat{A}_t^{(h)} = \frac{r_t^{(h)} - \mu_r}{\sigma_r},$$

where $\mu_r$ and $\sigma_r$ denote the mean and standard deviation of $r_t^{(h)}$ across the $G$ samples. Detailed designs and implementations for each reward component are provided in Appendix C.

**Low-Level Policy Optimization.** The low-level policy $\pi_\ell$ receives environment feedback based on whether it successfully completes the subgoal $g_t$, as defined by the environment reward $r_{\text{env}}(s_t, g_t, \pi_\ell)$ introduced above. For training, we reuse this signal as the step-level reward:

$$r_t^{(\ell)} = \begin{cases} 1 & \text{if } \pi_\ell \text{ completes } g_t, \\ 0 & \text{otherwise} \end{cases}, \quad \hat{A}_t^{(\ell)} = \frac{r_t^{(\ell)} - \mu_r^\ell}{\sigma_r^\ell}, \tag{4}$$

where $\mu_r^\ell$ and $\sigma_r^\ell$ denote the mean and standard deviation of $r_t^{(\ell)}$ across the current batch.

**Alternating Joint Optimization.** We alternate updates between $\pi_h$ and $\pi_\ell$ to facilitate coordination. At iteration $k$, we first fix $\pi_h^{(k-1)}$ and optimize $\pi_\ell^{(k)}$ using environment rewards, then fix $\pi_\ell^{(k)}$ and update $\pi_h^{(k)}$ with the foresight advantage:

$$\theta_\ell^{(k)} \leftarrow \arg\max_{\theta_\ell} \mathcal{J}_{\text{GRPO}}(\pi_\ell^{\theta_\ell} \mid \pi_h^{\theta_h^{(k-1)}}), \quad \theta_h^{(k)} \leftarrow \arg\max_{\theta_h} \mathcal{J}_{\text{GRPO}}(\pi_h^{\theta_h} \mid \pi_\ell^{\theta_\ell^{(k)}}). \tag{5}$$

### 4.3 Data Generation and Training Implementation

**Data Generation.** To enable efficient training, we construct an automated pipeline that interacts with Android emulators to generate subgoal-action trajectories. A hierarchical oracle—built from Qwen2.5-VL-72B (reasoning $\pi_h^*$) and Qwen2.5-VL-7B (action $\pi_\ell^*$)—produces demonstrations without manual annotation or rollbacks. To ensure a fair evaluation and mitigate data leakage, our process maintains a strict separation between training and test distributions. For AitW, we only reuse task instructions from the official splits to generate entirely new interaction trajectories, rather than using the original demonstration data. For the template-based AndroidWorld, we use different randomization seeds for the training and evaluation sets to prevent instance-level overlap. This process yielded over 1,200 high-quality, manually verified samples across all tasks. A comprehensive breakdown of our data construction protocol, dataset statistics, and a quantitative analysis of train-test overlap are provided in Appendix B.

Each trajectory $\tau = \{(s_t, u_t, \hat{g}_t, \hat{a}_t)\}_{t=1}^T$ consists of the UI screen state $s_t$, task instruction $u_t$, the generated semantic subgoal $\hat{g}_t$, and the corresponding atomic UI action $\hat{a}_t$, where:

$$\hat{g}_t \sim \pi_h^*(\hat{g} \mid s_t, u_t), \quad \hat{a}_t = \pi_\ell^*(\hat{a} \mid s_t, \hat{g}_t).$$

These trajectories serve as ground-truth references for computing the rewards described in Section 4.2, and are stored in structured JSON format:

```
{ "image_path":  "android/save/images/xxx.png",
"problem":  "Search for hotels in Washington DC",
"instruction":  "Click on the Chrome icon to open the browser.",
"solution":  { "action_type":  "DUAL_POINT",
"touch_point":  [0.7781, 0.6972] } }
```

**Training and Implementation.** We jointly train the high-level policy $\pi_h$ and the low-level policy $\pi_\ell$ using our modified GRPO scheme, which incorporates foresight advantage estimation and alternating optimization. Both components are instantiated with Qwen2.5VL-3B-Instruct.

Our GRPO-based training pipeline is implemented using the Huggingface TRL library[1] and the `GRPOTrainer` module from VLM-R1[2] (Shen et al., 2025). All experiments are conducted on four NVIDIA A800 80GB GPUs, with each training run taking approximately 22 hours. Complete implementation details, including data collection pipeline, training procedure, and model configuration, are provided in Appendix C.

---

[1] https://github.com/huggingface/trl
[2] https://github.com/om-ai-lab/VLM-R1

| | | AitW General | | WebShopping | |
|---|---|---|---|---|---|
| | | Train | Test | Train | Test |
| **Prompt-based** | SoM (GPT-4V) | 5.2 | 13.5 | 3.1 | 8.3 |
| | SoM (Gemini 1.5 Pro) | 32.3 | 16.7 | 6.3 | 11.5 |
| | AppAgent (GPT-4V) | 13.5 | 17.7 | 12.5 | 8.3 |
| | AppAgent (Gemini 1.5 Pro) | 14.6 | 16.7 | 5.2 | 8.3 |
| **Supervised Fine-tuned** | CogAgent | 25.0 | 25.0 | 31.3 | 38.5 |
| | AutoUI | 27.7 | 22.9 | 20.7 | 25.0 |
| | Filtered BC | $51.0 \pm 0.9$ | $54.5 \pm 1.3$ | $37.2 \pm 4.7$ | $43.8 \pm 1.7$ |
| **Reinforcement Learning** | Digi-RL | $63.5 \pm 0.0$ | $71.9 \pm 1.1$ | $68.2 \pm 6.8$ | $67.2 \pm 1.5$ |
| | Digi-Q | $61.5 \pm 2.3$ | $71.2 \pm 2.1$ | $53.1 \pm 1.7$ | $58.0 \pm 2.1$ |
| | **Hi-Agent (Ours)** | $\mathbf{76.4} \pm 0.2$ | $\mathbf{87.9} \pm 1.9$ | $\mathbf{70.3} \pm 0.2$ | $\mathbf{68.8} \pm 0.3$ |

Table 1: **Main comparisons on AitW benchmark.** Success rates (%) on the `General` and `WebShopping` subsets. Each RL-based method is run three times; mean and std are reported. Following prior work(Bai et al., 2024; 2025a), evaluation uses the first 96 instructions.

## 5 EXPERIMENTAL EVALUATION

We conduct a comprehensive evaluation of **Hi-Agent** on mobile device control tasks, focusing on four aspects: (i) task performance against prior baselines on the AitW benchmark (Section 5.1); (ii) generalization to unseen UI layouts and unseen tasks in Screenspot-v2 (Section 5.1); (iii) adaptability to different backbone models and training algorithms (Section 5.2); and (iv) scalability to larger models and more complex tasks on the AndroidWorld benchmark (Section 5.3).

**Environments.** *AitW* is a large-scale benchmark with five mobile control task categories(Rawles et al., 2023). Following prior work(Bai et al., 2024; 2025a), we evaluate on its two most challenging subsets—`General` and `WebShopping`—each consisting of the first 96 tasks. The former focuses on information access and app usage; the latter targets product search across e-commerce platforms.

**Observation and Action Space.** To ensure generalization, Hi-Agent operates under a unified observation and action space. Observations consist solely of RGB screenshots, without any structured UI annotations, bounding boxes, or Set-of-Marks (SoM) (Zheng et al., 2024). The action space includes normalized $(x, y)$ taps, long-presses, and swipes; variable-length text entry; functional button presses (e.g., `HOME`, `BACK`, `ENTER`); and task completion signals.

**Baselines.** We compare Hi-Agent against representative agents from three categories: *(1) Prompt-based agents*, which rely on large closed-source backbones (e.g., **GPT-4V** (OpenAI, 2024a), **Gemini 1.5 Pro** (Team et al., 2023)). We include **SoM** (Zheng et al., 2024) and **AppAgent** (Zhang et al., 2023) *(2) Supervised fine-tuned agents*, trained via imitation learning on labeled demonstrations with full parameter updates, including **CogAgent** (Hong et al., 2024), **AutoUI** (Zhang & Zhang, 2024), and **Filtered BC** (Pan et al., 2024). *(3) Post-trained RL agents*, optimized via offline or offline-to-online reinforcement learning. These agents directly update parameters based on task rewards. We include **DigiRL** (Bai et al., 2024) and **DigiQ** (Bai et al., 2025a).

### 5.1 MAIN PERFORMANCE AND GENERALIZATION ANALYSIS

**Task Performance on AitW. Hi-Agent** achieves **87.9%** and **68.8%** success rates on the `General` and `WebShopping` test sets, respectively, establishing a new SOTA. It surpasses the strongest prompt-based agents (APP Agent: 17.7% on `General`; SoM: 11.5% on `WebShopping`) by **+63.7%**, the best supervised method (Filtered BC: 54.5% on `General`; 43.8% on `WebShopping`) by **+29.2%**, and the top RL baseline (DigiRL: 71.9% on `General`; 67.2% on `WebShopping`) by **+8.8%**. These results highlight the benefits of our hierarchical design and foresight-guided optimization. We also identify environment errors in the original `WebShopping` setup—correcting them further boosts Hi-Agent's success rate to over 90%, as detailed in Appendix E.3.

**Analyzing Performance Gains.** To explain Hi-Agent's substantial gains over prior RL methods (e.g., +8.8% vs. DigiRL), we examine their failure modes. As shown in Figure 4a, over **70%** of both

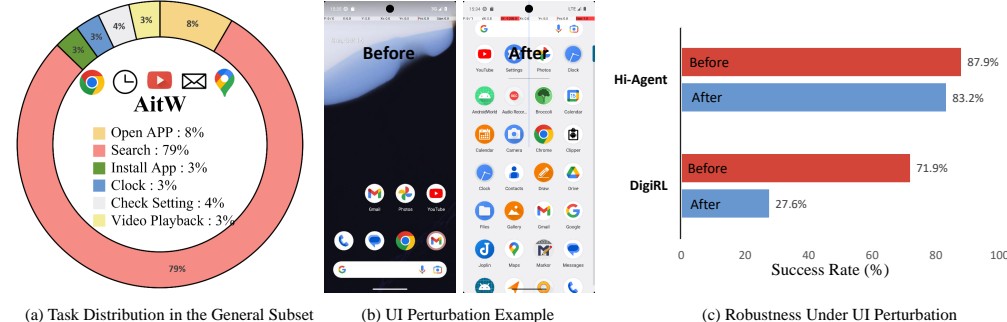

(a) Task Distribution in the General Subset     (b) UI Perturbation Example     (c) Robustness Under UI Perturbation

Figure 4: **Robustness analysis on AitW.** (a) Over 70% of General tasks are search-based, causing prior RL methods to overfit. (b) Layout shift from home screen to all-apps view alters app positions. (c) Hi-Agent remains robust (87.9% → 83.2%), while DigiRL drops sharply (71.9% → 27.6%).

training episodes and test tasks in the `General` subset are search-based (e.g., "search the weather in Paris"), reflecting a strong distributional skew. End-to-end RL agents tend to overfit to these dominant UI patterns—such as clicking fixed coordinates to launch Chrome and enter queries—while struggling to generalize to rare but structurally distinct tasks (e.g., "open Clock").

In contrast, Hi-Agent decouples reasoning and execution: the high-level model $\pi_h$ generates subgoals (e.g., "open Chrome"), while the low-level model $\pi_\ell$ grounds them into UI actions. This abstraction promotes skill reuse and generalization. We visualize representative success and failure cases in Appendix D, and provide a detailed analysis of task-wise performance in Appendix E.

**Robustness and Generalization.** We test Hi-Agent's generalization capabilities through two challenging scenarios. First, to assess robustness against UI layout perturbations, we change the agent's starting screen in AitW from the familiar home view to the all-apps view (Figure 4b). While DigiRL's performance drops sharply from **71.9%** to **27.6%**, exposing its reliance on memorized coordinates, Hi-Agent remains highly effective, with its success rate only dropping slightly from **87.9%** to **83.2%** (Figure 4c). The generalization capabilities of our architecture extend to the component level; our low-level action model ($\pi_\ell$), when trained on AitW, achieves competitive zero-shot performance on the ScreenSpot-v2 UI grounding benchmark (Wu et al., 2024). We provide detailed performance tables for the zero-shot evaluation in Appendix E.4 and qualitative visualizations of the layout perturbation experiment in Appendix E.

### 5.2 COMPONENT ABLATION AND ADAPTATION STUDY

We conduct ablation and adaptation studies on the AitW benchmark to assess the effectiveness and flexibility of our hierarchical framework.

**Ablation on Hierarchical Structure and Post-training.** We conduct an ablation study using Qwen2.5VL-3B as the backbone. We compare three configurations: (1) **Hi-Agent w/o Hierarchy & Post-train** (Qwen-3B (Raw)): the base model without hierarchy or training; (2) **Hi-Agent w/o Post-train** (Qwen-3B + Hierarchy): a two-level model with hierarchical structure but without post-training; (3) **Hi-Agent**: our full method with hierarchical decomposition and post-training.

As shown in Figure 5(a), incorporating hierarchy alone boosts performance from 1.6% to 60.0%, and full post-training further improves it to 87.9%, confirming the complementary benefits of task decomposition and RL-based post training. Appendix E.1 provides more details.

**Adaptation to Backbone Models.** To assess generalization to different base models, we replace Qwen2.5VL with GPT-4o and test it under two configurations: 1) **GPT-4o (Raw)**: the base model used directly without hierarchy; (2) **GPT-4o + Hierarchy**: augmented with our two-level structure, but without post training. As shown in Figure 5a, even without training, adding hierarchy improves GPT-4o's performance from 17.7% to 79.8%, demonstrating the general utility of our design.

**Comparison with Supervised Fine-Tuning.** We compare our RL-based approach against standard supervised fine-tuning (SFT) on the same hierarchical Qwen-3B architecture and training data, using

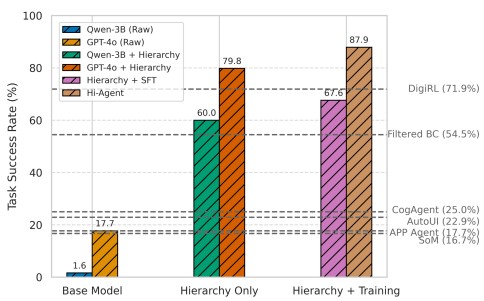 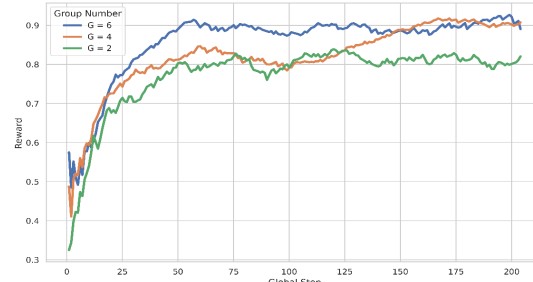

(a) Component Effectiveness and Adaptability      (b) Effect of Group Number $G$ on Training Stability

Figure 5: **Effectiveness and efficiency of Hi-Agent.** (a) Task success across model scales and training algorithms, showing consistent gains from hierarchical modeling and post-training. (b) Training curves under different group sizes $G$; larger $G$ improves stability and speeds up convergence.

LLaMA-Factory[3] for the SFT implementation. As shown in Figure 5(a), while SFT achieves a respectable 67.6% success rate, our GRPO-based Hi-Agent reaches a significantly higher 87.9%. This suggests our RL solution is more robust, promoting better generalization where SFT can overfit to demonstration patterns in dynamic GUI environments.

**Impact of Group Size in GRPO.** We further investigate the effect of the group size $G$ in our improved GRPO. Figure 5(b) shows training curves under different $G$ values. Larger groups yield more stable learning signals and faster convergence by providing better estimates of relative advantage. This confirms the practical importance of $G$ in balancing efficiency and robustness.

## 5.3 SCALABILITY TO LARGER MODELS AND MORE COMPLEX TASKS

To assess scalability, we evaluate **Hi-Agent** with larger models on the more challenging Android-World benchmark, which requires stronger reasoning, planning, and fine-grained control than AitW.

We scale both the high-level model $\pi_h$ and low-level model $\pi_\ell$ in Hi-Agent. As shown in Table 2, our hierarchical framework scales effectively with model size and consistently improves performance under greater task complexity. In particular, the configuration using a 72B reasoning model and a 7B action model achieves a 56.5% success rate, outperforming the GPT-4o baseline by over 22 absolute points (56.5% vs. 34.5%). A detailed per-task success breakdown and visual illustrations on Android-World are provided in Appendix E.5.

These results highlight that our method scales to high-capacity models and complex tasks. By decoupling reasoning and execution, Hi-Agent enables large models to generalize better and solve long-horizon tasks efficiently.

Table 2: AndroidWorld task success rates. *denotes post-trained models.

| Model | Success Rate |
|---|---|
| Qwen2-VL-2B (fine-tuned) | 9.0 |
| GPT-4 Turbo (Rawles et al., 2024) | 30.6 |
| GPT-4o (Wang et al., 2024b) | 34.5 |
| GPT-4o + UGround (Gou et al., 2024) | 44.0 |
| GPT-4o + Aria-UI (Yang et al., 2024) | 44.8 |
| UI-TARS (Qin et al., 2025) | 46.6 |
| Agent S2 (Agashe et al., 2025) | 54.3 |
| **Hi-Agent (3B***+**3B***)** | 26.3 |
| **Hi-Agent (7B***+**7B***)** | 31.9 |
| **Hi-Agent (32B+7B***)** | 43.9 |
| **Hi-Agent (72B+7B***)** | **56.5** |

## 6 CONCLUSION

We propose **Hi-Agent**, a scalable hierarchical vision-language agent that decouples high-level subgoal reasoning and low-level action execution. By combining structured task decomposition with foresight-guided GRPO optimization, Hi-Agent significantly outperforms prompt-based, supervised, and RL-based baselines in both task success and generalization, while maintaining strong scalability with model size and task complexity.

---

[3]https://github.com/hiyouga/LLaMA-Factory

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

APPENDIX

**Statement on LLM Usage.** In preparing this manuscript, Large Language Models (LLMs) were employed solely as auxiliary tools for improving linguistic quality. Their role was restricted to refining grammar, enhancing readability. At no point did LLMs contribute to the conception of research ideas, the design of methods, the execution of experiments, or the interpretation of results. All scientific content and conclusions are entirely the work of the authors.

**Appendix Overview.** The supplementary material provides additional theoretical analysis, implementation details, and extended experimental results to support the main paper.

- Section A presents a formal analysis of global optimality under recursive decomposition in our hierarchical framework.
- Section B describes our data construction protocol and provides a detailed analysis of the train-test overlap to ensure fair evaluation.
- Section C details the training procedure, including data generation, model configurations, and hyperparameter settings.
- Section D provides qualitative case studies, including both successful and failure examples to illustrate model behavior.
- Section E includes extended experiments: evaluation under UI layout perturbation, a zero-shot generalization test on the ScreenSpot-v2 benchmark, analysis of the WebShopping subset, and additional statistics on the AndroidWorld benchmark.

Our code and data are included in the supplementary material to support full reproduction.

## A  GLOBAL OPTIMALITY VIA RECURSIVE CONSTRUCTION

Here, we formally establish the conditions under which a recursively optimal hierarchical policy $(\pi_h, \pi_\ell)$ achieves global optimality. Following the notation and recursive decomposition structure defined in Section 4.1, we consider an MDP $M$ hierarchically decomposed into subtasks $\{M_0, M_1, \ldots, M_m\}$, with $M_0$ representing the root task.

**Proposition 1** (Global Optimality Condition). *Let $\pi^*$ denote the optimal flat policy for MDP $M$. Assume this optimal sequence can be partitioned into a sequence of valid subtasks under the hierarchical decomposition. Then, a recursively optimal hierarchical policy $\pi = (\pi_h, \pi_\ell)$ is also a globally optimal policy, i.e., $V^\pi(s_0) = V^{\pi^*}(s_0)$.*

*Proof.* We prove by contradiction. Assume that the recursively optimal policy $\pi$ is *not* globally optimal. This implies there exists another hierarchical policy $\tilde{\pi}$ such that for some starting state $s_0$, its value is strictly greater: $V^{\tilde{\pi}}(s_0) > V^\pi(s_0)$.

Let us identify the first decision point $(s_k, M_k)$ where the policies diverge. At this state, $\pi$ chooses subgoal $g_k$ while $\tilde{\pi}$ chooses a different subgoal $g'_k$. Since this is the first point of divergence, the value obtained by following $\tilde{\pi}$ from this state onward must be strictly greater than that from following $\pi$.

However, a recursively optimal policy $\pi$, by definition, selects the subgoal that maximizes the expected future return. This return is captured by the hierarchical Q-value:

$$Q_k^\pi(s_k, g) = V_g^\pi(s_k) + C_k^\pi(s_k, g).$$

The completion function $C_k^\pi(s_k, g)$ correctly accounts for stochastic termination by averaging over the distribution of all possible exit states and durations, as defined in Eq. (2).

The choice made by the recursively optimal policy $\pi$ at state $s_k$ is therefore:

$$g_k = \arg\max_g Q_k^\pi(s_k, g).$$

A direct consequence of this maximization is that for any alternative subgoal $g'_k$, the following inequality must hold:

$$Q_k^\pi(s_k, g_k) \geq Q_k^\pi(s_k, g'_k).$$

This implies that switching the choice from $g_k$ to $g'_k$ cannot increase the expected value from state $s_k$ onward. This contradicts our earlier deduction that the value of policy $\tilde{\pi}$ (which chose $g'_k$) must be strictly greater.

Therefore, our initial assumption that $\pi$ is not globally optimal must be false. Hence, a recursively optimal hierarchical policy is globally optimal. $\qquad\square$

**Expressivity.** Our two-level hierarchical framework consists of a high-level reasoning policy $\pi_h$ generating semantic subgoals $g_t$, and a low-level policy $\pi_\ell$ executing these subgoals via primitive actions $a_t$. Given that $\pi_h$ can directly emit atomic actions as subgoals, and $\pi_\ell$ is capable of executing them, the joint policy space $(\pi_h, \pi_\ell)$ fully encompasses the space of flat policies. Therefore, recursively optimal hierarchical policies retain the expressivity necessary for achieving global optimality.

**Foresight Advantage.** To further align local subgoal optimization with global task success, we introduce a foresight advantage:

$$\hat{A}_t^{(h)} = \frac{r_t^{(h)} - \mu_r}{\sigma_r}, \quad \text{where} \quad r_t^{(h)} = \lambda_1 r_{\text{fmt}}(g_t) + \lambda_2 r_{\text{env}}(s_t, g_t, \pi_\ell) + \lambda_3 \hat{V}_{\text{judge}}(s_t, g_t).$$

Here, $r_{\text{fmt}}$ reflects syntactic and semantic subgoal correctness, $r_{\text{env}}$ evaluates execution feedback from the environment, and $\hat{V}_{\text{judge}}$ estimates long-term subgoal feasibility via a pretrained VLM oracle. This reward shaping mechanism mitigates the risk of locally greedy yet globally suboptimal subgoal selection, guiding $\pi_h$ to reason with foresight and converge toward globally optimal task strategies.

# B  DATA CONSTRUCTION AND OVERLAP ANALYSIS

We carefully avoided data leakage between training and evaluation. Below, we clarify the data preparation and task partitioning across both AitW and AndroidWorld benchmarks.

**AitW (General & WebShopping)** We selected the first 96 instruction texts from the official splits—matching baseline evaluation setups—and generated new trajectories using our automatic data collection pipeline (Section 4.3). We did not use any raw trajectories from the original dataset; only instruction texts were reused. All collected trajectories were manually verified for correctness.

**AndroidWorld** This benchmark uses parameterized task templates such as "Create a new contact for {name} with number {number}". Each task instance is dynamically generated with randomized parameters. We ensured that training and evaluation used different random seeds to avoid any template-level overlap.

**Task Overlap Quantification** We provide a detailed quantification of task overlap between training and evaluation sets in Table 3. The minor overlap in AitW stems from a small number of tasks that appear in both the original train and test splits provided by the benchmark creators, for which we used the instruction texts. Our methodology ensures no trajectory-level overlap.

Table 3: Task overlap analysis between training data generation and evaluation sets.

| Benchmark | #Tasks Used in Training | #Tasks in Test | Overlap Ratio |
|---|---|---|---|
| AitW-General | 96 | 96 | 6.25% |
| AitW-WebShopping | 96 | 96 | 5.21% |
| AndroidWorld | 116 | 116 | 0% |

**Dataset Scale** We provide detailed statistics of our training data across all benchmarks in Table 4. For AitW, we selected 96 task instructions from the training splits of both `General` and `WebShopping`, consistent with prior work. We re-executed each using our Oracle agent, collecting new trajectories. After manual verification and filtering, this resulted in **205 verified samples**

for AitW-General and **389 for AitW-WebShopping**. For AndroidWorld, which defines 116 param-eterized task templates, we instantiated one randomized goal per template and collected training samples via the Oracle policy. We retained **682 high-quality samples** after manual validation.

Table 4: Training data statistics across all benchmarks.

| Benchmark | Subset | #Task Instructions | #Verified Samples |
|---|---|---|---|
| AitW | General | 96 | 205 |
| AitW | WebShopping | 96 | 389 |
| AndroidWorld | Full | 116 | 682 |

## C  DETAILED TRAINING PIPELINE

Our hierarchical training framework decomposes long-horizon mobile tasks into single-step sub-goals, enabling efficient optimization using GRPO. Here, we elaborate on the full training pipeline, emphasizing the mechanisms used to generate training signals and their integration within GRPO.

### C.1  REWARD DATASET GENERATION

Due to limitations of the Android emulator regarding state rollback, obtaining rewards by sequen-tially interacting with the environment becomes computationally expensive. Therefore, we design an oracle model based on our hierarchical architecture, consisting of a Qwen2.5-VL-72B reason-ing model paired with a Qwen2.5-VL-7B action model, to automatically generate accurate reward datasets without manual labeling. To ensure high-quality data generation, we carefully crafted prompts for the 72B reasoning model, guiding it to generate reliable subgoal instructions condi-tioned on task descriptions, previous actions, and current screen states.

For clarity, we present the exact prompt structure used by the 72B reasoning model below:

```
You are a mobile operation Agent that performs precise screen interactions.  Analyze the
input and generate the next action instruction.

# Task Description

Execute multi-step mobile tasks through sequential single-step decisions.

# Input Components

{ "image":  "Screen image (analyze UI elements)",
"text/Previous Actions":  ["action_type":  "...", "touch_point":  "[x,y]", ...],
"text/Goal":  "Current task objective" }

# Action Output Components

You should only output concise and clear action instructions, including action types and
action targets, without specific coordinates.

# Output Format (strictly follow):

<reasoning>
1.  Analyze previous action sequence
2.  Identify important elements in current screen
3.  Determine required next-step action instruction
</reasoning>

<instruction>
Instruction:  ...
</instruction>

#Examples
example1:
Input:  "Previous Actions:  xx
Output:  <reasoning> xx </reasoning> <instruction> xx </instruction>
example x:  xxx
```

We empirically verify the oracle model's effectiveness on the first 96 tasks from the AitW bench-mark. The hierarchical oracle, powered by the structured prompt and dual-model architecture, achieves a task success rate of 93.2%. This demonstrates that our oracle can reliably serve as an

automated annotator for large-scale subgoal-action data collection, enabling scalable and accurate training.

An example of the collected reward dataset is presented below, structured clearly in JSON format for consistency and ease of use:

```
{
"image_path":  "android/save/images/test3/1743001445.7178237_1.png",
"problem":  "Search for hotels in Washington DC",
"instruction":  "Click on the Chrome icon to open the browser.",
"solution":  "Action Decision: {action_type:  DUAL_POINT, touch_point:  [0.7781, 0.6972],
lift_point:  [0.7781, 0.6972], typed_text:  ""
}
```

During training, the `"solution"` field serves as a reference signal for both the low-level action model and the high-level reasoning model: the former receives direct execution supervision, while the latter is optimized via foresight rewards that incorporate oracle feedback on the executability and quality of predicted subgoals.

## C.2 GRPO TRAINING PROCEDURE

We leverage the constructed reward dataset to post-train both components of our hierarchical policy using a modified GRPO framework. To train the high-level reasoning model $\pi_h$, we compute a foresight reward signal by aggregating three components: format reward, execution feedback reward, and subgoal feasibility reward. Each is described below.

**Format Reward.** To ensure subgoals generated by the reasoning model conform to a syntactically valid and semantically interpretable structure, we define a binary format reward $r_{\text{fmt}}(g_t)$ based on regular expression matching. Only subgoals matching the following format receive positive reward:

```
<reasoning> ...  </reasoning>
<instruction>Instruction:  ...</instruction>
```

This pattern ensures that each subgoal contains both a reasoning trace and a structured instruction. Subgoals that omit either tag or violate the structural layout are penalized with zero format reward.

**Execution Feedback Reward.** To supervise the low-level action model $\pi_\ell$, we compare its predicted action $a_t = \pi_\ell(s_t, g_t)$ against the oracle action $\hat{a}_t$ in the dataset. The reward $r_{\text{env}}(s_t, g_t, a_t)$ is defined as:

$$r_{\text{env}}(s_t, g_t, a_t) = \mathbb{1}\left\{\text{type}(a_t) = \text{type}(\hat{a}_t) \wedge \|\text{coord}(a_t) - \text{coord}(\hat{a}_t)\|_2 < \epsilon\right\},$$

where $\epsilon$ is a threshold for coordinate similarity (set to $0.002$ in our experiments). In the formula, $\text{type}(a_t)$ and $\text{type}(\hat{a}_t)$ respectively denote the action types of $a_t$ and $\hat{a}_t$, while $\text{coord}(a_t)$ and $\text{coord}(\hat{a}_t)$ correspond to the coordinates of the actions $a_t$ and $\hat{a}_t$. This reward is also propagated to the high-level model to encourage generation of executable subgoals.

**Subgoal Feasibility Reward.** To measure whether the predicted subgoal $g_t$ is appropriate and feasible under the current screen context, we employ a frozen Qwen2.5-VL-72B model as a subgoal evaluator. The evaluation prompt is carefully designed to enforce atomicity, context validity, keyboard preconditions, and target presence. Only when all criteria are satisfied is a reward of 1 returned; otherwise, the reward is 0. The exact prompt used is as follows:

```
You are a mobile operation instruction validator.  Strictly evaluate if the generated
instruction is valid for the current step.

# Evaluation Criteria (ALL must be met for reward=1):
1.  Atomic Action:  Must represent ONE actionable step (e.g., "click X" not "click X then
do Y")
2.  Context Match:  Must logically follow from the current screen state
3.  Keyboard State:  For text input instructions, keyboard MUST be visible/activated
4.  Target Existence:  Referenced UI element must be present in current screen

# Evaluation Rules:
- Reward=1 ONLY when ALL criteria are satisfied
- Reward=0 for ANY violation

# Examples:
```

```
[Valid Example 1]
Task:  Search for hotels in Washington DC
Screen:  Home screen with Chrome icon
Instruction:  "click Chrome"
→ Reward=1

[Invalid Example 1]
Task:  Search for hotels in Washington DC
Screen:  Chrome search page (no keyboard)
Instruction:  "type hotels"
→ Violates Rule 3 → Reward=0

[Valid Example 2]
Task:  Search for hotels in Washington DC
Screen:  Chrome search page (keyboard visible)
Instruction:  "type hotels in Washington DC"
→ Reward=1

[Invalid Example 2]
Task:  Search for hotels in Washington DC
Screen:  Chrome search page (no keyboard)
Instruction:  "click search bar"
→ Valid action → Reward=1

[Invalid Example 3]
Task:  Search for hotels in Washington DC
Screen:  Home screen
Instruction:  "open Chrome and search"
→ Violates Rule 1 → Reward=0

[Edge Case]
Task:  Search for hotels in Washington DC
Screen:  Search results page
Instruction:  "click back button"
→ Valid but unrelated to task → Still Reward=1

# Output Format:
{"reward":  1} or {"reward":  0}
NO explanations.  Strict JSON format only.
```

This evaluator forms the third term in the foresight reward:

$$r_t^{(h)} = \lambda_1 \cdot r_{\text{fmt}}(g_t) + \lambda_2 \cdot r_{\text{env}}(s_t, g_t, \pi_\ell) + \lambda_3 \cdot \hat{V}_{\text{judge}}(s_t, g_t)$$

This integrated signal guides the reasoning model to produce subgoals that are both well-formed and pragmatically executable, closing the loop between semantic intent and environmental grounding.

## C.3  TRAINING AND DEPLOYMENT OF HIERARCHICAL MODELS

Following the construction of the reward dataset, we proceed to train a compact reasoning model using the oracle-generated subgoal-instruction pairs. To promote better generalization and minimize manual prompt engineering, we intentionally adopt an extremely simplified prompt for the 3B-scale reasoning model. This design choice ensures that the model can generalize beyond prompt-specific templates and reduces deployment complexity. The same prompt is used for both training and inference:

```
You are a mobile operation Agent that performs precise screen interactions.  Analyze the
input and generate the next action instruction.
STRICTLY follow this structure:
<reasoning> reasoning process here </reasoning> <instruction>Instruction:
...</instruction>
```

**Low-Level Execution via Function-Call Interface.** The low-level action model $\pi_\ell$ interacts with the Android environment through structured API-based function calls. Each atomic action is expressed as a JSON-formatted tool invocation, providing clear semantics for device control. The exact prompt used is as follows:

```
# Tools

You may call one or more functions to assist with the user query.
The following function is available:
```

```
<tools>
{
"name": "mobile_use",
"description": "Use a touchscreen to interact with a mobile device, and take screenshots.
The screen's resolution is 1092x2408. Supported actions include clicking, typing,
swiping, system button presses, and more.",
"parameters": {
"action": ["click", "type", "swipe", "key", "system_button", "terminate"],
"coordinate": [x, y],
"text": "Optional input text",
"button": ["Back", "Home", "Menu", "Enter"],
"status": ["success", "failure"]
}
}
</tools>
```

This interface allows the high-level reasoning model to focus exclusively on intent prediction, while the low-level action model translates these into executable atomic operations. It not only simplifies control flow but also improves modularity and debugging during large-scale mobile task execution.

## C.4 TRAINING HYPERPARAMETERS AND CONFIGURATION

We adopt the GRPOTrainer implementation from VLM-R1[4] (Shen et al., 2025) for training our high-level and low-level models. The complete training configuration is summarized in Table 5.

| Hyperparameter | Value |
|---|---|
| Max Prompt Length | 526 |
| Number of Generations ($G$) | 6 |
| Batch Size per Device | 3 |
| Gradient Accumulation Steps | 2 |
| Number of Training Epochs | 2 |
| Max Completion Length | 256 |
| Optimizer Precision | `bfloat16` |
| Gradient Checkpointing | `true` |
| Attention Implementation | `flash_attention_2` |
| Temperature | 0.9 |
| Top-$p$ | 1.0 |
| Top-$k$ | 50 |
| Repetition Penalty | 1.0 |
| Learning Rate | $1 \times 10^{-6}$ |
| KL Coefficient ($\beta$) | 0.04 |
| Foresight Reward Weight $\lambda_1$ | 0.4 |
| Foresight Reward Weight $\lambda_2$ | 0.3 |
| Foresight Reward Weight $\lambda_3$ | 0.3 |
| Clipping Threshold ($\epsilon$) | 0.2 |

Table 5: Training hyperparameters used for hierarchical model optimization.

Training is conducted on four NVIDIA A800 80GB GPUs, and each full run takes approximately 22 hours to complete. The key software stack includes `flash_attn` 2.7.4.post1, `torch` 2.6.0, `transformers` 4.49.0, and `trl` 0.16.0.dev0. These configurations ensure stable training, efficient memory usage via FlashAttention, and compatibility with the GRPOTrainer pipeline.

## C.5 BASELINE HYPERPARAMETERS AND CONFIGURATION

We employed a series of baseline models, setting their hyperparameters in strict accordance with the configurations reported in the original papers. The training configuration is summarized in Table 6.

Table 6 presents the primary training hyperparameters for DigiRL (Bai et al., 2024) and DigiQ (Bai et al., 2025a). For a more comprehensive list of settings, please refer to the original papers.

---

[4]https://github.com/om-ai-lab/VLM-R1

| Method | Hyperparameter | Value |
|--------|----------------|-------|
| DigiRL | actor lr | 3e-3 |
| | value function lr | 3e-3 |
| | instruction value function lr | 3e-3 |
| | batch size | 128 |
| | rollout trajectories | 16 |
| | replay buffer size | 5000 |
| | rollout temperature | 1.0 |
| | maximum gradient norm | 0.01 |
| | GAE $\lambda$ | 0.5 |
| | actor updates per iteration | 20 |
| | value function updates per iteration | 5 |
| | instruction value function updates per iteration | 5 |
| Digi-Q | actor lr | 1e-4 |
| | value function lr | 1e-5 (general), 5e-6 (webshop) |
| | batch size | 128 |
| | maximum gradient norm | 0.01 |
| | actor updates per iteration | 30 |
| | value function updates per iteration | 20 |

Table 6: Baseline Methods Hyperparameters

## D  CASE STUDY: QUALITATIVE ANALYSIS OF HI-AGENT

We present qualitative examples to visualize the hierarchical reasoning and action execution process of Hi-Agent. The goal is to provide insight into how the agent decomposes abstract task instructions into semantic subgoals, and grounds them into executable atomic actions on the mobile UI. As evidenced by the examples, the agent now exhibits a clear capacity both to interpret given instructions and to manipulate the smartphone interface.

### D.1  ILLUSTRATION OF TASK DECOMPOSITION

Figure 6 shows an example where Hi-Agent successfully completes the task *"Send a message to Alice"*. The reasoning model decomposes the goal into subgoals such as opening Messenger, selecting the receiver box, and typing the contact name. These semantic subgoals are then executed through low-level UI actions (e.g., `Click(x,y)`, `Input "Alice"`), bridging symbolic reasoning and visual grounding.

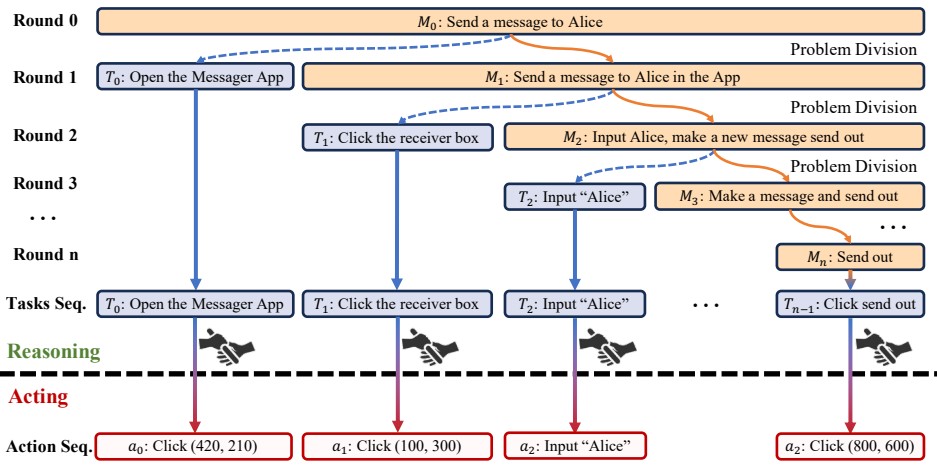

Figure 6: Hi-Agent first decomposes the high-level task into interpretable subgoals, then executes them via grounded UI actions.

## D.2 QUALITATIVE SUCCESS EXAMPLES

Figures 7 and 8 depict procedural tasks involving the Clock application, which require structured interaction across multiple UI states.

In Figure 7, the agent completes the task *"Open the clock"* by identifying the correct application icon from the home screen or app drawer and issuing a click command. Although visually simple, this case tests the agent's ability to robustly locate app-specific UI elements under varying layouts. Figure 8 demonstrates a more complex interaction: *"Set an alarm for 4PM"*. The reasoning model first identifies that this goal entails a sequence of subtasks—launching the Clock app, selecting the "Alarm" tab, configuring the time selector to "4" and "PM," and confirming the alarm setup. The action model grounds these semantic instructions into a precise sequence of atomic UI operations. This example showcases the hierarchical policy's capacity to parse abstract temporal goals and execute interface-specific configurations through multi-step navigation.

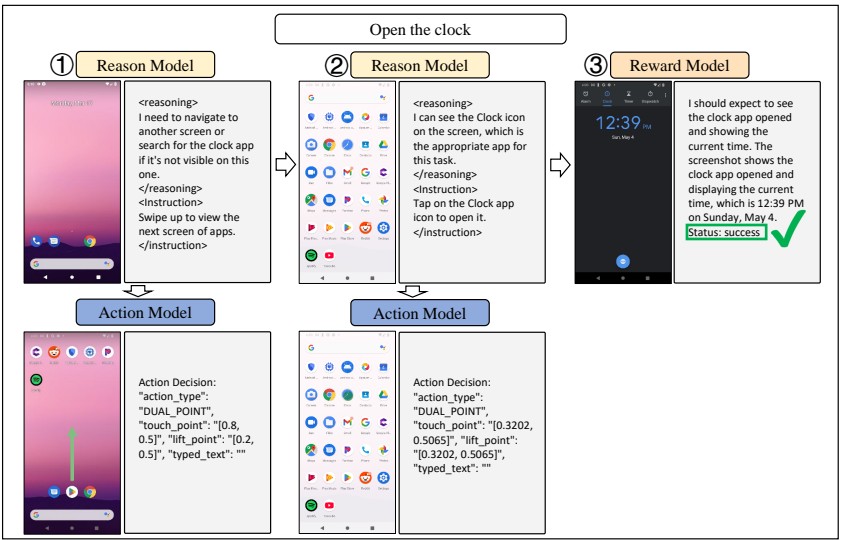

Figure 7: Illustration of Hi-Agent completing the task *"Open the clock"*.

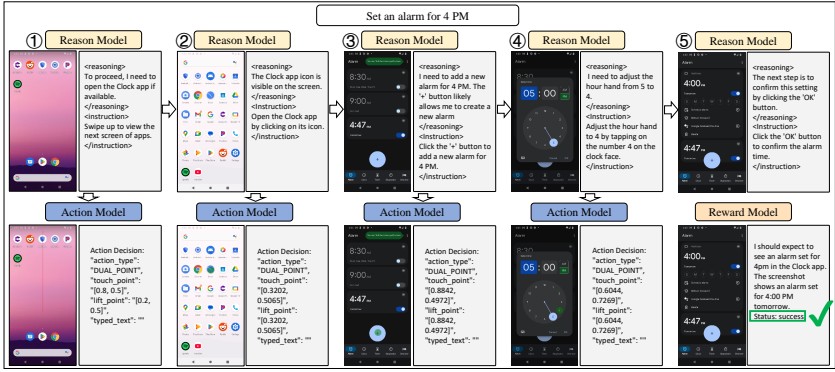

Figure 8: Illustration of Hi-Agent completing the task *"Set an alarm for 4PM"*.

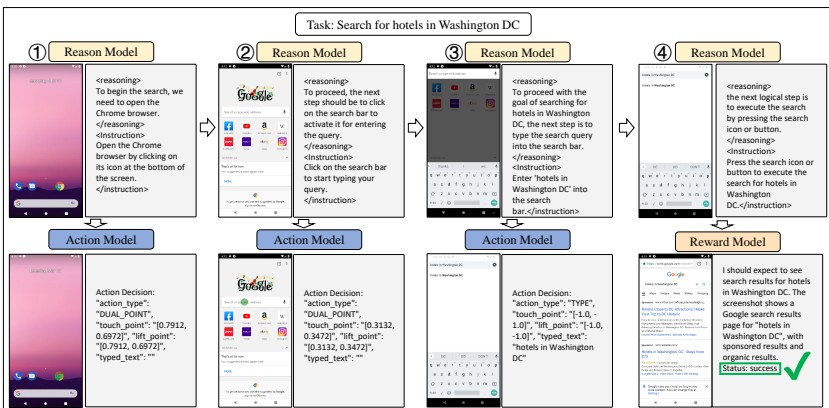

Figure 9: Illustration of Hi-Agent completing the task *"Search for hotels in Washington DC"*.

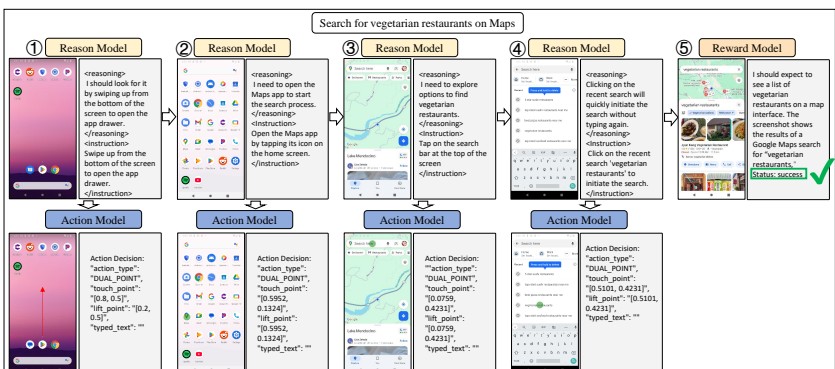

Figure 10: Illustration of Hi-Agent completing the task *"Search for vegetarians on Maps"*.

Figures 9–11 present search-oriented scenarios, demonstrating Hi-Agent's capability to interpret user intents, select appropriate applications, and execute tasks via step-wise decomposition. In Figure 11, the directive "Play the new Drake video on YouTube" is fulfilled by launching YouTube, issuing a text query, parsing the result list, and selecting the most recent entry—illustrating content retrieval, ambiguity resolution, and UI grounding. Figure 9 depicts a Chrome-based web search for "hotels in Washington DC," where the agent opens Chrome, inputs the query, and awaits the search results, thus emulating standard browser workflows. Figure 10 shows the task "Search for vegetarians on Maps," in which the agent launches Maps, activates the search bar, and issues a location-based query, evidencing spatial reasoning and semantic grounding.

Across these examples, Hi-Agent demonstrates the ability to plan multi-step routines, recover from intermediate states, and generate semantically appropriate and executable instructions under diverse application contexts.

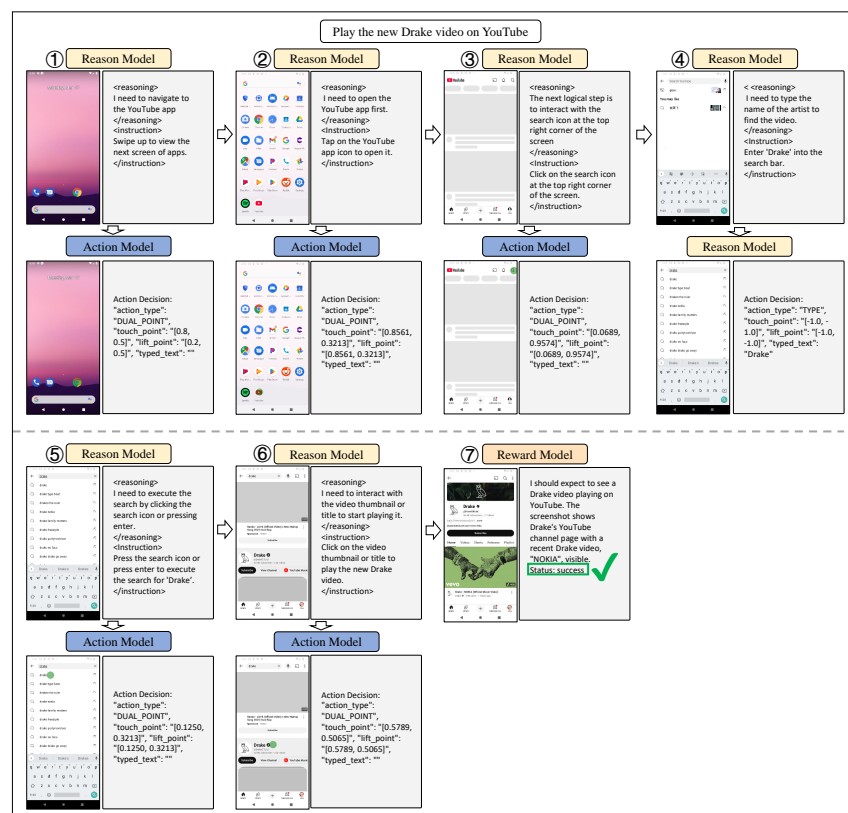

Figure 11: Illustration of Hi-Agent completing the task *"Play the new Drake video on YouTube"*.

### D.3 FAILURE TAXONOMY AND ANALYSIS

We categorize common failure cases into five representative types observed across evaluation tasks:

**Complex UI or Missing Target:** the required UI element is ambiguous or absent (e.g., an out-of-stock product or visually occluded item), leading to redundant actions (repeated swipes) and step-overflow.

**External Dependency or Latency:** slow page loads, missing content, or emulator instability cause premature termination, since our action space lacks a dedicated "wait" operation.

**Incorrect Navigation Path:** the agent selects an unintended app or menu (e.g., using a third-party app to access system settings), resulting in irrecoverable divergence from the optimal path.

**Premature Termination:** the agent exits before achieving the final subgoal, often due to reward misalignment or incorrect completion assumption.

**Goal Misunderstanding:** partial misinterpretation of instructions (e.g., confusing search with shopping tasks or opting for web search instead of app interaction).

Figures 12 and 13 illustrate two failure cases representative of the first two categories. Figure 12 illustrates a task failure that occurred when the website lacked the cargo required to complete the mission. Figure 13 depicts how network instability combined with constraints on the action space led to repeated access attempts and ultimately caused the step-count limit to be exceeded.

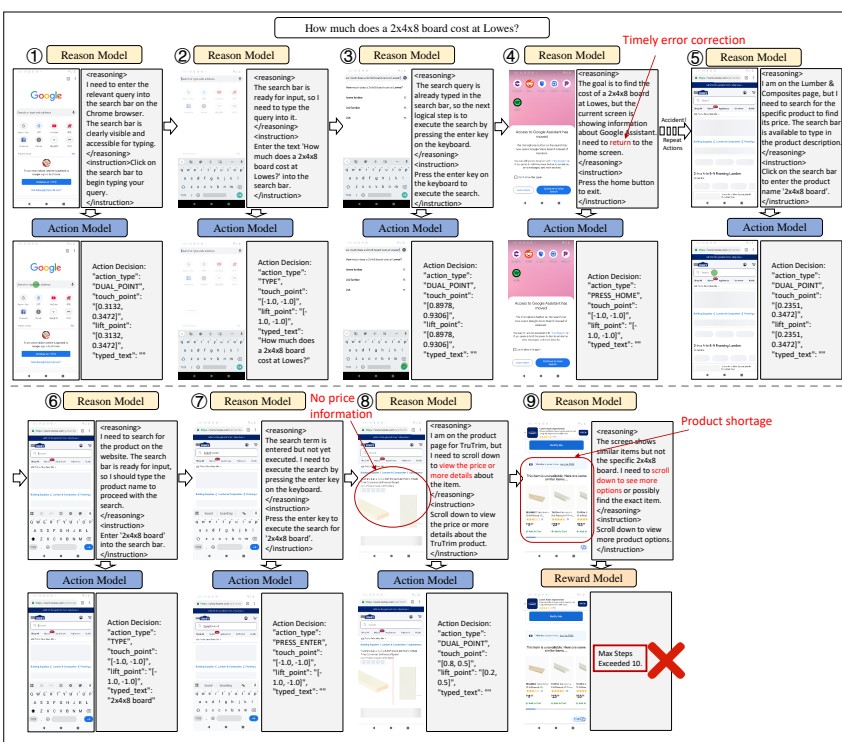

Figure 12: Failure due to product unavailability and repeated swiping exceeding the step limit.

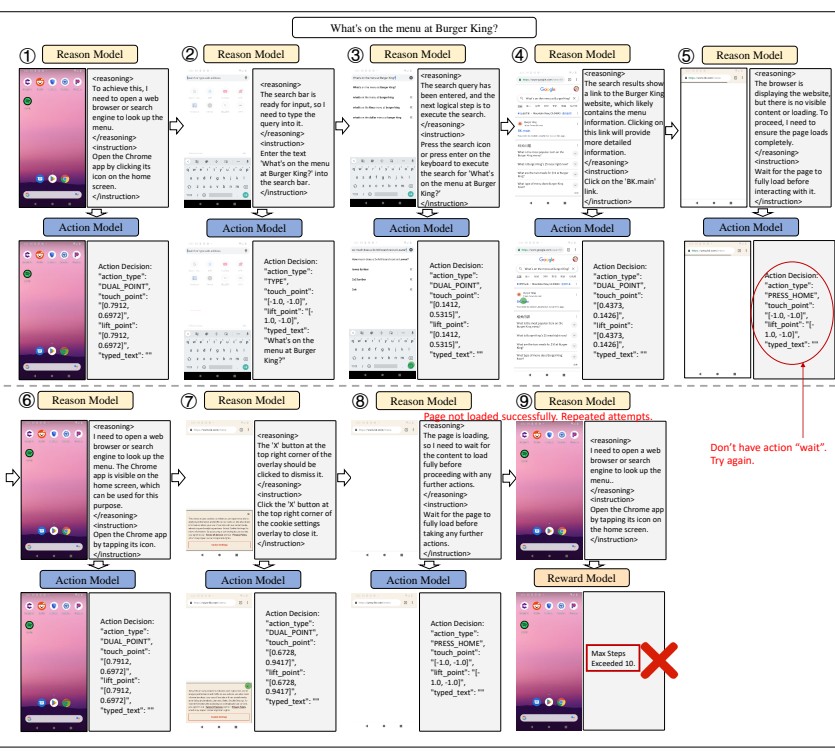

Figure 13: Failure due to long page load and the lack of a "wait" operation.

# E ADDITIONAL RESULTS

This section provides complementary analyses and visualizations for four extended analyses beyond the main evaluation: (1) distribution of task completion outcomes, including the proportion of each error category after classification, (2) generalization under UI layout shift, (3) error diagnosis and correction for WebShopping tasks in AitW, and (4) large-scale deployment of Hi-Agent in the AndroidWorld benchmark.

## E.1 FAILURE TASKS DISTRIBUTION

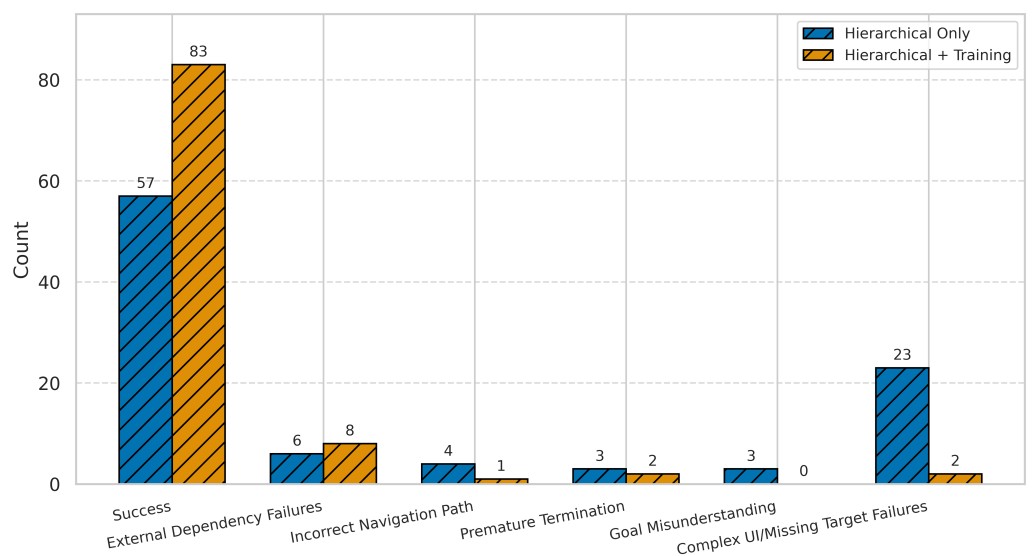

Figure 14: Distribution of task success and failure across all evaluation instances.

As shown in Figure 14, after GRPO training, the hierarchical model exhibits a remarkable enhancement in its comprehension of application pages and a noticeable improvement in task disassembly. Consequently, the task success rate of the hierarchical model following GRPO training has significantly increased. The rise in errors related to External Dependency or Latency is attributed to the previous model failing to access the correct website and encountering task failures before facing network issues. After the model's capability was elevated through GRPO training, these inherent environmental issues were laid bare.

## E.2 ROBUSTNESS TO LAYOUT PERTURBATION

We visualize how layout changes impact agent performance in Figure 15. The task is *"What's a good restaurant in Las Vegas?"*. During training, agents are initialized from the home screen, but in this evaluation setting, the starting screen is changed to the all-apps view, causing a significant layout shift. Under this condition, DigiRL fails to locate Chrome and instead opens the Contacts app, repeating incorrect actions. In contrast, Hi-Agent successfully completes the task. Thanks to its hierarchical architecture, the high-level reasoning model remains unaffected by coordinate-level changes and generates consistent subgoals, while the low-level action model grounds those subgoals to new visual contexts.

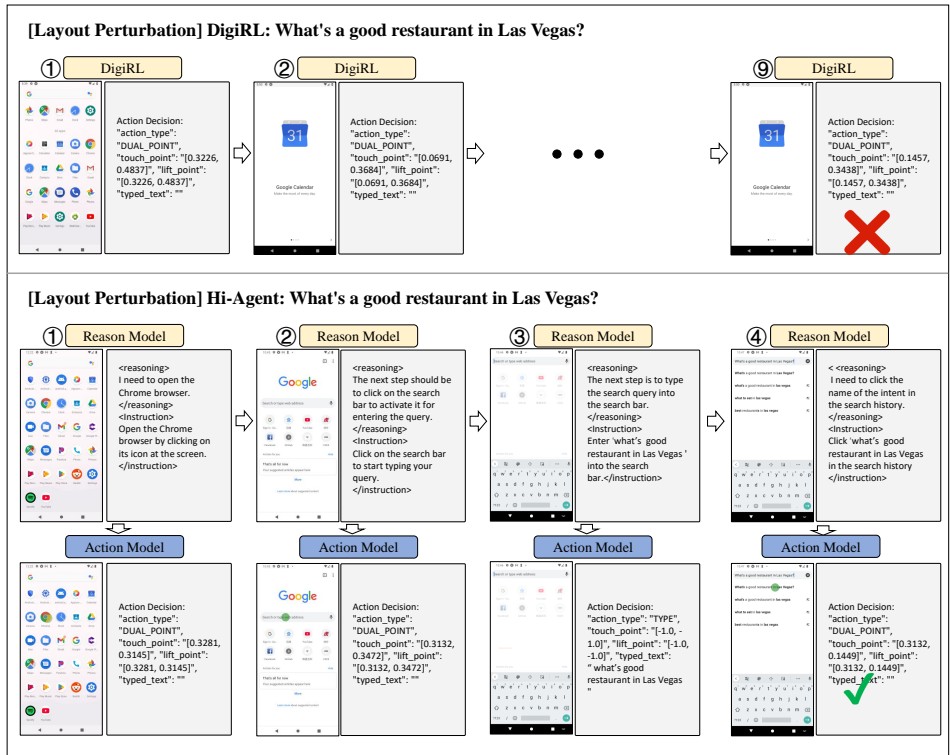

Figure 15: Layout shift visualization for the task *"What's a good restaurant in Las Vegas?"*. When the layout is perturbed by switching from the home screen (training setting) to the all-apps view (test setting), DigiRL fails to locate Chrome and repeatedly interacts with the wrong app. In contrast, Hi-Agent completes the task successfully by leveraging its hierarchical decomposition, which enables robust subgoal generation and grounding under spatial variation.

### E.3 ERROR ANALYSIS AND CORRECTION ON AITW WEBSHOPPING SUBSET

We analyze the agent's performance on the AitW WebShopping task subset. The current success rates are 70.3% on training and 68.8% on testing tasks. After manual inspection, we find that two problematic domains—`newegg.com` and `costco.com`—consistently lead to failure: the former blocks agent access, while the latter prevents `<ENTER>` key inputs. This observation aligns with prior findings reported in DigiQ (Bai et al., 2025a). When we replace these domains with `ebay.com` and rerun the `WebShopping` subset, success rates improve significantly to 92.7% on train and 91.2% on test (see Table 7).

Table 7: Success rate before and after WebShopping subset correction.

|  | **Original** | **After Domain Replacement** |
|---|---|---|
| Train Subset | 70.3% | **92.7%** |
| Test Subset | 68.8% | **91.2%** |

### E.4 ZERO-SHOT GENERALIZATION ON GUI GROUNDING

To further validate the generalization capability of our hierarchical architecture, we evaluate the low-level action model ($\pi_\ell$) on the **ScreenSpot-v2 benchmark** (Wu et al., 2024) in a zero-shot setting. ScreenSpot-v2 is a comprehensive GUI grounding benchmark spanning mobile, web, and desktop platforms, designed to test an agent's fundamental ability to locate text and icon/widget elements. For this experiment, we take the low-level model trained on AitW data and directly apply it to ScreenSpot-v2 without any fine-tuning.

As shown in Table 8, our 7B low-level action model achieves a **highly competitive 91.5% average score** in this zero-shot setting, outperforming several larger, specialized SFT models. This result demonstrates that our training framework encourages the action model to learn robust and generalizable visual representations, rather than merely overfitting to the training tasks. The model's strong grounding ability across diverse platforms is further illustrated by the qualitative examples in Figure 16.

Table 8: Zero-shot performance on the **ScreenSpot-v2** benchmark. Our low-level model ($\pi_\ell$) is evaluated without any fine-tuning on this dataset. Baselines are from original papers.

| Models | Mobile | | Desktop | | Web | | Avg |
|---|---|---|---|---|---|---|---|
| | Text | Icon | Text | Icon | Text | Icon | |
| *Closed-source Models* | | | | | | | |
| GPT-4o | 26.6 | 24.2 | 24.2 | 19.3 | 12.8 | 11.8 | 20.1 |
| UI-TARS-1.5 | - | - | - | - | - | - | 94.2 |
| Seed1.5-VL | - | - | - | - | - | - | 95.2 |
| *GUI-specific Models (SFT)* | | | | | | | |
| SeeClick-9.6B | 78.4 | 50.7 | 70.1 | 29.3 | 55.2 | 32.5 | 55.1 |
| UGround-7B | 75.1 | 84.5 | 85.1 | 61.4 | 84.6 | 71.9 | 76.3 |
| UI-TARS-7B | 96.9 | 89.1 | 95.4 | 85.0 | 93.6 | 85.2 | 91.6 |
| Jedi-7B | 96.9 | 87.2 | 95.9 | 87.9 | 94.4 | 84.2 | 91.7 |
| GUI-Actor-7B | 97.6 | 88.2 | 96.9 | 85.7 | 93.2 | 86.7 | 92.1 |
| *GUI-specific Models (RL)* | | | | | | | |
| UI-R1-E-3B | 98.2 | 83.9 | 94.8 | 75.0 | 93.2 | 83.7 | 89.5 |
| LPO | 97.9 | 82.9 | 95.9 | 86.4 | 95.6 | 84.2 | 90.5 |
| GTA1-7B | 99.0 | 88.6 | 94.9 | 89.3 | 92.3 | 86.7 | 92.4 |
| GTA1-72B | 99.3 | 92.4 | 97.4 | 89.3 | 95.3 | 91.4 | 94.8 |
| *Ours (Zero-Shot from AitW)* | | | | | | | |
| **Hi-Agent** ($\pi_\ell$, 7B) | 96.6 | 81.0 | 95.9 | 84.3 | 94.9 | 91.1 | 91.5 |

## E.5 LARGE-SCALE DEPLOYMENT ON ANDROIDWORLD

In Section 5.3, we show that Hi-Agent scales to larger models and more complex mobile environments. Using a Qwen2.5-VL-72B reasoning model and a 7B action model, our hierarchical agent achieves a success rate of 56.5% on the AndroidWorld benchmark—demonstrating competitive performance among methods that rely solely on raw screenshots as input.

Figure 18 illustrates Hi-Agent solving a structured input task: *"Add the following expenses into the pro expense: Movie Night—375.45—Entertainment—Urgent"*. The agent opens the expense app, fills each field accurately, and uses swiping gestures to select the correct category. Figure 19 shows a temporal query task: *"What is on my schedule for October 28 at 2:45am in Simple Calendar Pro?"*. The agent distinguishes between multiple dates on the UI (e.g., 28 at the top vs. bottom), selects the correct one, and parses event information directly from the screen.

We also report per-app success statistics in Figure 17. The agent performs reliably on structured apps such as Clock, Settings, and Expense. In contrast, it struggles with apps like Markor and Retro, which require prior usage familiarity—without such user-specific guidance, even humans may find them hard to operate. Another common failure mode involves tasks lacking explicit termination signals (e.g., taking a photo, scrolling to the end of a page). Unlike humans, who adjust behavior dynamically through feedback, the agent only receives discrete visual frames, making it hard to infer when the task should be terminated.

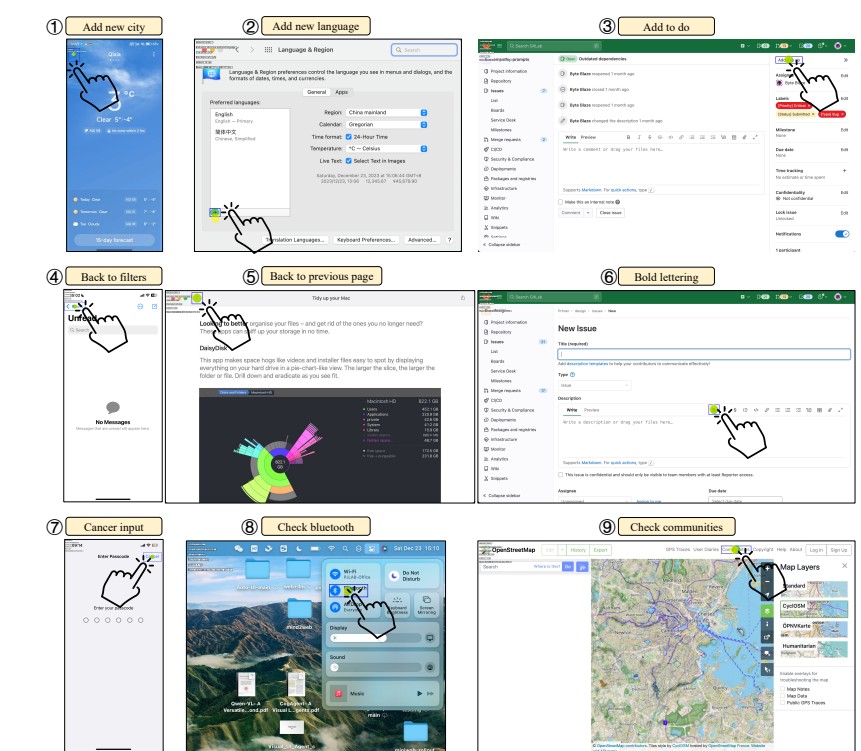

Figure 16: Qualitative examples of Hi-Agent's zero-shot GUI grounding performance on diverse tasks from the ScreenSpot-v2 benchmark.

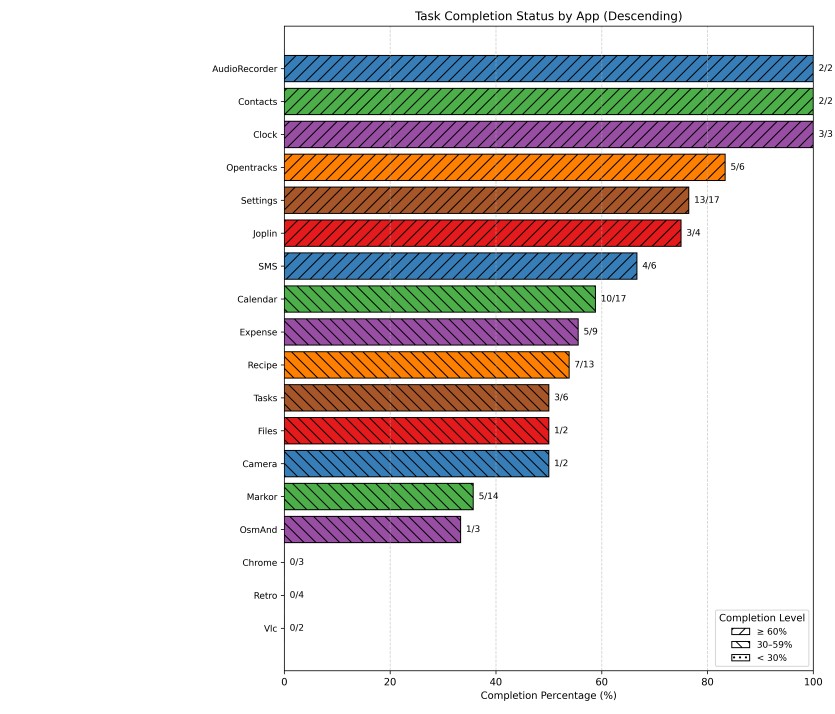

Figure 17: Per-app success statistics in AndroidWorld.

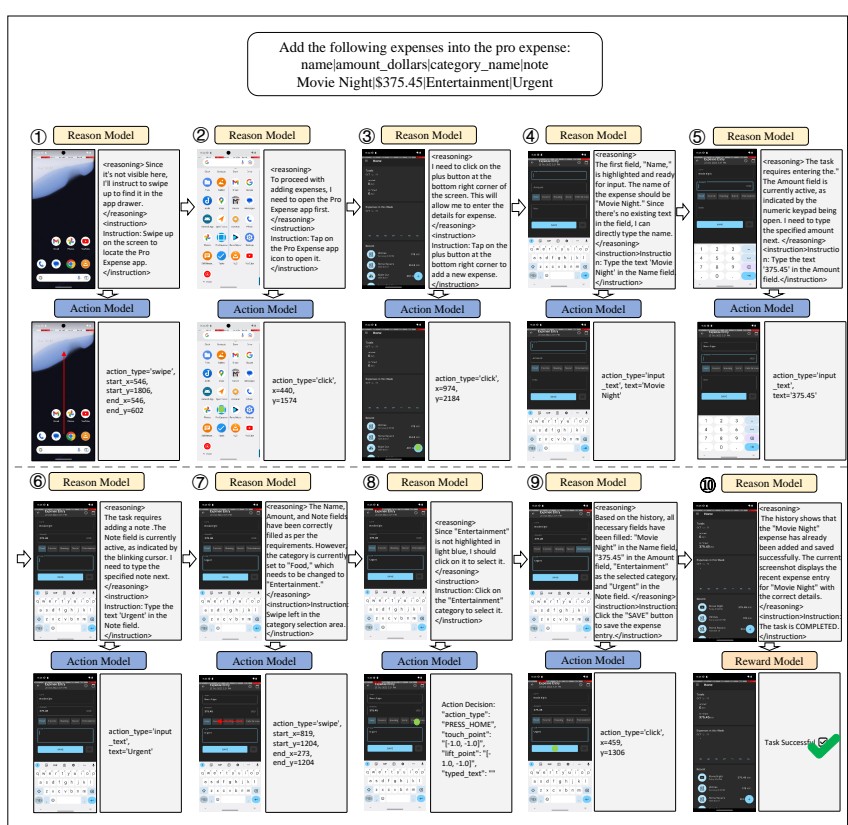

Figure 18: Illustration of Hi-Agent completing the task *"Add the following expenses into the pro expense: name|amount_dollars|category_name|note Movie Night|$375.45|Entertainment|Urgent"*.

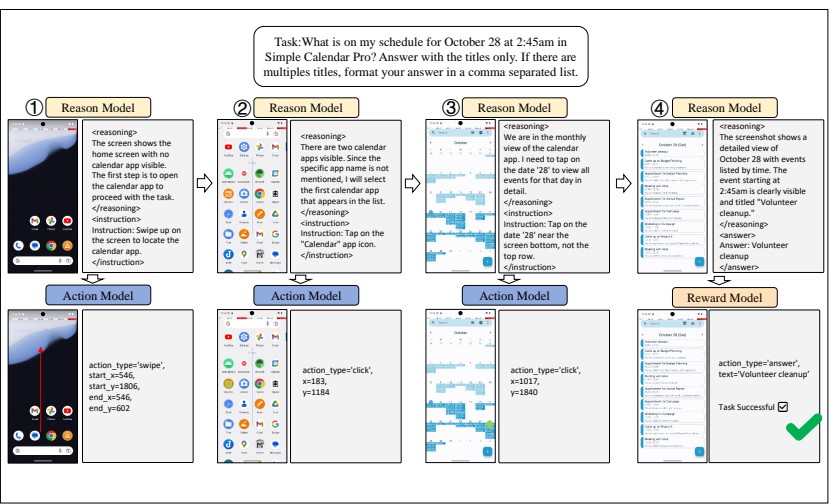

Figure 19: Illustration of Hi-Agent completing the task *"What is on my schedule for October 28 at 2:45am in Simple Calendar Pro? Answer with the titles only. If there are multiples titles, format your answer in a comma separated list. "*.

