# OpenReview forum: "Hi-Agent: Hierarchical Vision-Language Agents for Mobile Device Control"
_ICLR.cc/2026/Conference — ICLR 2026 Conference Withdrawn Submission_

### Official Review · Reviewer_xriS · 2025-10-27

**Soundness:** 2
**Presentation:** 3
**Contribution:** 3
**Rating:** 2
**Confidence:** 3

**Summary:**

The paper proposes a two-stage hierarchical multi-agent structure for GUI agent. It splits the common used one-model GUI agent reasoning and action process to two distinct agents and provides a RL training framework to reduce the trajectory path explosion.

**Strengths:**

- Separating the overall task to subtasks and reducing the path computation is a novel idea to overcome the possibility explosion
- Though the idea of multi-agent is already widely used in many frameworks, finetuning multiple agents is a good idea to overcome the limitation of single model.

**Weaknesses:**

The most weaknesses are in the experiments.
1. AitW is almost abandoned by main-stream researches. Almost no SOTA models evaluate on AitW, basically due to its significant drawbacks such as wrong annotations. Also the authors only evaluate on 96 tasks in two categories. The full evaluation subset in AitW contains 5 categories and much more tasks. The evaluation setting on AitW can hardly demonstrate the effectiveness. For the static evaluation, AndroidControl is a better choice.
2. AndroidWorld is a representative dynamic benchmark, but authors only compare with commercial general models (GPT + different models), UI-TARS and Agent S2. Recently, before the submission deadline, several more single-model agent are released such as UI-TARS-2.0, UI-Venus and GUI-OWL, and they all achieve much higher scores with smaller-size models. In Table 2, 72B + 7B achieves 56.5 while other single-model 7B agents can achieve >60%, which questions whether using the two-stage multi-model agent is better?
3. The training data collection seems biased. Authors directly collect trajectories from AitW and AndroidWorld tasks and they train and evaluate on the same set of template and instruction text as the evaluation set, which seems the only overfits to those tasks. What about other tasks in AitW evaluation? What about generalization to tasks in AndroidLab? The improper data collection makes me question the effective of the training paradigm

To summarize, the experiments and results can't demonstrate the effectiveness of the agent structure design and the training paradigm

**Questions:**

1. how does the training data used? Are the data collected in both AitW and AndroidWorld mixed in training or each split is trained separately and evaluated separately

---

### Official Review · Reviewer_4nT7 · 2025-10-28

**Soundness:** 2
**Presentation:** 2
**Contribution:** 2
**Rating:** 4
**Confidence:** 5

**Summary:**

This paper introduces Hi-Agent to address the limitations of existing agents that rely on direct state-to-action mappings, which often lack structured reasoning and poor generalization. The core innovation is a hierarchical architecture with a high-level reasoning model (for generating subgoals) and a low-level action model, which are jointly optimized. Hi-Agent achieves an 87.9% task success rate on the Android-in-the-Wild benchmark, significantly outperforming prior methods, while also showing strong zero-shot generalization on ScreenSpot-v2 and effective scalability on the complex AndroidWorld benchmark.

**Strengths:**

* Hi-Agent framework allows both the high-level reasoning model and low-level action model to be updated simultaneously.
* The author conduct experiment on diverse benchmarks, and Hi-Agent achieves state-of-the-art performance.

**Weaknesses:**

* Some statements in the paper may be misleading.
  1. In Figure 1, the paper’s work appears to obtain rewards from environmental feedback, whereas in reality the authors use offline-collected trajectories as ground truth, and construct feedback rewards based on the consistency between predictions and the ground-truth steps. This is consistent with earlier methods (e.g., UI-R1 [1]).
  2. In Section Training and Implementation, the authors describe the training resource overhead but do not indicate the model size of the high-level policy model. Evidently, this is not the 72B model mentioned earlier.
  3. Since the subtask decomposition granularity of HI-Agent is at the step level, it seems equivalent to directly training two independent models with separate training objectives at the step level. Why do the authors call this joint training?
* The construction of training data may introduce unfair factors: the authors collect ground-truth trajectories in real environments using benchmark instructions for HI-Agent, whereas the compared models or agent frameworks often use open-sourced data or operate in a zero-shot setting. Therefore, the experiments cannot demonstrate that the proposed method has better generalization capability.
* The proposed method has already appeared in prior work. For example, GTA1 [2] uses o3 as the reasoner and GTA-1 as the executor. UI-S1 [3] performs reinforcement learning based on open-sourced trajectories. The authors do not sufficiently discuss the differences between their method and these existing approaches.

[1] Lu, Zhengxi, et al. "UI-R1: Enhancing Efficient Action Prediction of GUI Agents by Reinforcement Learning." arXiv preprint arXiv:2503.21620 (2025).

[2] Yang, Yan, et al. "Gta1: Gui test-time scaling agent." arXiv preprint arXiv:2507.05791 (2025).

[3] Lu, Zhengxi, et al. "UI-S1: Advancing GUI Automation via Semi-online Reinforcement Learning." arXiv preprint arXiv:2509.11543 (2025).

**Questions:**

* Did the authors use all trajectories for training, or train different models with corresponding data for each benchmark? This question arises from the note “Zero-Shot from AitW” in Table 8 of the appendix — does this mean that the evaluation at that point is of a model trained on AitW data?
* What are the prompts used for grounding evaluation and for the low-level action model, respectively?

---

### Official Review · Reviewer_p93w · 2025-10-30

**Soundness:** 4
**Presentation:** 4
**Contribution:** 4
**Rating:** 8
**Confidence:** 4

**Summary:**

This work proposes a novel algorithm for developing GUI agents, featuring a hierarchical structure (using a high-level planner and a low-level actor) with joint optimization. To effectively optimize the agents, the authors propose a variant of GRPO that alleviates discrepancies between environment-level MDP and token-level MDP and the exponential growth of trajectories, causing excessive computational costs. As a result, the trained agent demonstrates the state-of-the-art results in Android-in-the-Wild and promising scores competitive to proprietary baselines in AndroidWorld. Additional investigations related to scaling and failures are also discussed, allowing a deeper understanding.

**Strengths:**

I provide the strengths of this paper.
1. Promising results: above all, the results are promising. State-of-the-art results in AitW and results that are competitive to proprietary models (e.g., GPT-4o) in AndroidWorld with 72B+8B models (as well as probing the potential growth in scaling) are remarkable achievements.
2. Novel method: the introduction of a training method (based on GRPO) for tasks with long horizons is very desirable, and the actual realization is also notable.
3. Behavior analysis: analysis of where the gains stemmed from is also very reasonable. It provides a general insight into why the proposed method could demonstrate superior results compared to the baselines, also hinting at the next directions of future algorithms (i.e., what is solved and what would still be missing). Component examination is presented neatly, resolving questions on the designs.

**Weaknesses:**

I present several weaknesses/questions/suggestions here.
1. Improving AndroidWorld results: are the authors willing to test by scaling more than 72B+7B? I agree that the current set-up already shows a promise, a combination of Qwen2.5-VL-72B (reasoning) + Qwen3-VL-8B (action) is worth a try until the final revision. Hopefully, we might be able to observe something more interesting.
2. Adding more references: I provide more related work, focusing on training methods [1,2] and more benchmarks [3,4,5,6], worth including for providing a broader landscape to the readers. I note that testing the proposed methods on the aforementioned benchmarks does not seem highly demanding, as the results are already based on standardized benchmarks.
3. Details for equation (1): for readability, concretely define what each ‘compositive’ and ‘primitive means. Also, why P & R are conditioned in i? Does this mean that the rewards are provided differently with respect to the value of i?
4. Baseline performance: I am curious about the performance of the Hi-agents before training. What are the specific values of them (i.e., at training step = 0 in Fig.5-b, if my understanding is correct)?
5. Latency of Hi-Agent: From Fig.14 (in Appendix E.5), I assume that Hi-Agent requires more delays between steps. How long does it take to succeed in a task (on average)? If it’s possible, I’d like to also see comparisons with other baselines, as I believe that the latency is a crucial issue for the practicality of GUI agents.

---

References:

[1] Wang et al., “DistRL: An Asynchronous Distributed Reinforcement Learning Framework for On-Device Control Agents” (2024).

[2] Papoudakis et al., “AppVLM: A Lightweight Vision Language Model for Online App Control” (2025).

[3] Zhang et al., “Mobile-Env: Building Qualified Evaluation Benchmarks for LLM-GUI Interaction” (2023).

[4] Lee et al., “Benchmarking Mobile Device Control Agents across Diverse Configurations” (2024).

[5] Zhang et al., “LlamaTouch: A Faithful and Scalable Testbed for Mobile UI Task Automation” (2024).

[6] Chen et al., “SPA-Bench: A Comprehensive Benchmark for SmartPhone Agent Evaluation” (2024).

**Questions:**

(Please see above)

---

### Official Review · Reviewer_f9mN · 2025-11-03

**Soundness:** 3
**Presentation:** 1
**Contribution:** 2
**Rating:** 4
**Confidence:** 5

**Summary:**

This paper proposes a hierarchical modeling approach for device control agents. Here, two policies are jointly optimized: a high-level reasoning policy, and a lower-level action policy. The proposed approach achieves SOTA performance on an Android device control task and competitive performance with other approaches on other device control benchmarks.

**Strengths:**

* Decomposing the problems of high-level reasoning and low-level action execution for embodied tasks is well-motivated.
* The proposed approach performs well on Android device control benchmarks.

**Weaknesses:**

* The intro has a lot of citations; these should mostly be reserved for related work unless crucial for the point being made.
* In terms of writing, I think there needs to be more refinement in terms of formalization. There is a lot of formalization, but it isn't adding much to the paper, in my opinion. For example, all of section 4.1 is spent on defining value function decomposition, but then these formalisms are never used again. Instead, I am confused on the distinction, if there is one, between whether the architecture is hierarchical, recursive, or both. It's not clear, for example, in Figure 2, what V and C refer to. What is the actual input/output space of the reasoning model, and the action model? My main point: there are a lot of formal details (e.g., the exact JSON format for storing training data), but there are inconsistencies among them, and they distract from the bigger picture. This is making it difficult for me to evaluate the core contributions of the proposed framework (beyond the framework itself existing).

**Questions:**

* Why would GRPO result in an exponential number of rollouts? Please answer with respect to how data is sampled during training (inputs and outputs to the reasoner/action models). I don't understand what is combinatorial about the sampling process.
* Section 4.3 defines a process for creating 1,200 samples of training data from oracle policies. Are the policies being trained actually used for online sampling during training?

---

### Note · Authors · 2025-12-03

I have read and agree with the venue's withdrawal policy on behalf of myself and my co-authors.